# Functional Annotation of Human Cognitive States using Graph Convolution Networks

**Yu Zhang, and Pierre Bellec**
Department of Psychology, Université de Montréal, Montreal, QC H1X 2B2
Centre de recherche de l'Institut universitaire de gériatrie de Montréal, Montreal, QC H3W 1W6
`yuzhang2bic@gmail.com; pierre.bellec@gmail.com`

## Abstract

A key goal in neuroscience is to understand brain mechanisms of cognitive functions. An emerging approach is to study "brain states" dynamics using functional magnetic resonance imaging (fMRI). So far in the literature, brain states have typically been studied using 30 seconds of fMRI data or more, and it is unclear to which extent brain states can be reliably identified from very short time series. In this project, we applied graph convolutional networks (GCN) to decode brain activity over short time windows in a task fMRI dataset, i.e. associate a given window of fMRI time series with the task used. Starting with a populational brain graph with nodes defined by a parcellation of cerebral cortex and the adjacent matrix extracted from functional connectome, GCN takes a short series of fMRI volumes as input, generates high-level domain-specific graph representations, and then predicts the corresponding cognitive state. We investigated the performance of this GCN "cognitive state annotation" in the Human Connectome Project (HCP) database, which features 21 different experimental conditions spanning seven major cognitive domains, and high temporal resolution task fMRI data. Using a 10-second window, the 21 cognitive states were identified with an excellent average test accuracy of 89% (chance level 4.8%). As the HCP task battery was designed to selectively activate a wide range of specialized functional networks, we anticipate the GCN annotation to be applicable as a base model for other transfer learning applications, for instance, adapting to new task domains.

## 1 Introduction

Identifying brain networks involved in human cognition has been one of the main goals of neuroscience research. Modern imaging techniques, such as functional magnetic resonance imaging (fMRI), provide an opportunity to accurately map the neural substrates of human cognition. An emerging topic in the literature is the identification of "brain states", characterized by a canonical spatial pattern of functional activity, which were found to associate with specific cognitive states. A popular approach to identify these brain states, called multi-voxel pattern analysis (MVPA), uses machine learning tools to decode which task a subject performed based on recordings of brain activity in task fMRI (8). But the algorithm is usually limited to specific cognitive domains and relies on long acquisition of brain activity with repeated blocks to accurately decode a brain state.

In this project, we proposed a GCN architecture for annotating human brain activity on a cognitive battery of 21 task states. Instead of using the averaging BOLD signals or statistical constrast maps, GCN takes a short series of fMRI volumes as input, generates task-specific graph representations, and then predicts the corresponding cognitive labels. Comparing to the multi-class support vector machines classification, GCN achieved much higher performance in identifying a variety of cognitive states.

33rd Conference on Neural Information Processing Systems (NeurIPS 2019), Vancouver, Canada.

## 2 Datasets and Preprocessing

In this project, we are using block-design task fMRI data from the Human Connectome Project (HCP) S1200 release. The minimal preprocessed fMRI data of the CIFTI format were used, which maps individual fMRI time series onto the standard surface template with 32k vertices per hemisphere. Further details on fMRI data acquisition, task design and preprocessing can be found in (2) and (5).

The task fMRI data includes seven cognitive tasks, which are emotion, gambling, language, motor, relational, social, and working memory. In total, there are 23 different cognitive states. Considering the short event design nature of the gambling trials (1.5s for button press, 1s for feedback and 1s for ITI), we evaluated the decoding models with and without the two gambling conditions and found a much lower precision and recall scores for gambling task (average f1-score = 61%) than other cognitive domains (average f1-score > 91%). In the following experiments, we excluded the two gambling conditions and only reported results on the remaining 21 cognitive states. The detailed description of the tasks can be found in (2). A summary table is also shown in Tab. 1.

Table 1: Parameters of task-designs from HCP dataset.

| Task domain | Subjects | Runs | Cond | Volumes per run | Trials per run | Minimal Dura(sec) |
|---|---|---|---|---|---|---|
| Working Memory | 1085 | 2 | 8 | 405 | 10 | 25 |
| Motor | 1083 | 2 | 5 | 284 | 8 | 12 |
| Language | 1051 | 2 | 2 | 316 | 8 | 12 |
| Social Cognition | 1051 | 2 | 2 | 274 | 5 | 23 |
| Relational Processing | 1043 | 2 | 2 | 232 | 6 | 16 |
| Emotion Processing | 1047 | 2 | 2 | 176 | 6 | 18 |

## 3 Graph Signal Processing and Graph Convolution Network

Starting with a brain signal $x$, Graph signal processing (GSP) first maps the signal onto a weighted graph $\mathcal{G} = (\mathcal{V}, \mathcal{E}, W)$ that defines a network structure among a set of brain regions. The set $\mathcal{V}$ is a parcellation of cerebral cortex into $N$ regions, and $\mathcal{E}$ is a set of connections between each pair of brain regions, with its weights defined as $W_{i,j}$. Here we used the multimodal cortical parcellation of the cerebral cortex (4), which delineates 180 functional areas per hemispheree bounded by sharp changes in cortical architecture, function, connectivity, and topography. The connections between brain areas were estimated by calculating the group averaged resting-state functional connectivity (RSFC) based on 1080 minimal prepossessed resting-state fMRI data (5). The spectral analysis of the graph signal relies on the graph Laplacian, defined as:

$$L = I - D^{-1/2}WD^{-1/2}, \tag{1}$$

where $D$ is a diagonal matrix of node degrees and $I$ is the identity matrix. As we assume the weights to be undirected and symmetric, the matrix $L$ can be factored as $U\Delta U^T$, where $U = (u_0, \ldots, u_{N-1})$ is the matrix of Laplacian eigenvectors, also called graph Fourier modes, and $\Delta = \text{diag}(\lambda_0, \ldots, \lambda_{N-1})$ is a diagonal matrix with the corresponding eigenvalues, specifying the frequency of the modes. This eigendecomposition can be interpreted as a generalization of the standard Fourier basis onto a non-Euclidean domain. Thus, the graph Fourier transform is defined as $\hat{x} = \mathcal{L}\{x\} = U^T x$ and its inverse as $x = U\hat{x}$, where $x$ is the graph signal.

Merging the spectral graph theory with deep learning techniques, (3) first proposed a new architecture of graph convolutional neural networks (GCN), which generates a linear combination of the graph modes across the full spectrum of graph Laplacian. Kipf and colleagues (7) introduced a simplified version of GCN:

$$x * g_\theta = \theta(I + D^{-1/2}WD^{-1/2})x, \tag{2}$$

where $\theta$ is a single parameter to be learned. Thus, the output of a graph convolution layer is defined as:

$$X^{l+1} = \sigma(\widetilde{W}X^l\Theta^l), \quad \widetilde{W} = I + D^{-1/2}WD^{-1/2} \tag{3}$$

where $X^l \in \mathbb{R}^{N \times F}$ denotes the matrix of graph signals on layer $l$, with $N$ brain regions and $F$ graph filters. $\Theta^l \in \mathbb{R}^{F_{in} \times F_{out}}$ is the parameters to be learned on layer $l$ with $F_{in}$ income channels/filters

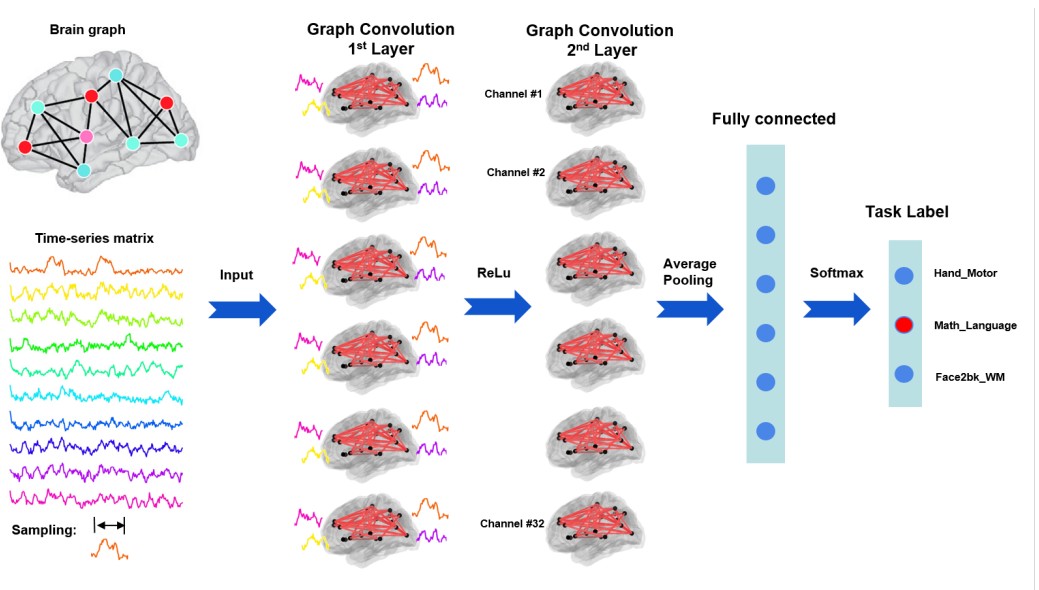

Figure 1: Pipeline of functional brain decoding using graph convolutions

and $F_{out}$ outcome filters. These parameters are shared among all nodes on layer $l$. $\sigma(.)$ denotes an activation function, such as the ReLU$(x) = \max(0, x)$. It's worth noted that this type of graph convolution only takes into account the direct neighbours in the graph. By stacking multiple GCN layers, we could propagate information of brain activity among $k_{th}$-order neighbourhood, with $k$ being the number of GCN layers. Specifically, in the first GCN layer, we treat the short series of fMRI volumes as multiple channels, with $X^1 \in \mathbb{R}^{N \times F}$ being a 2D matrix of $N$ brain regions and $F$ time windows. During model training, the first GCN layer learns various versions of the spatio-temporal convolution kernels (spatially only consider the direct neighbours, and temporally use the same size as the time window), as a replacement of the canonical hemodynamic response function (HRF). The derived activation maps are then inputted to the following convolutional layers to generate high-level graph representations.

A brain state annotation model was proposed, consisting of 6 GCN layers with 32 graph filters at each layer, followed by a global average pooling layer and 2 fully connected layers. The model takes short series of fMRI data as input, propagates information among inter-connected brain regions and networks, generates a high order graph-level representations and finally predicts the corresponding cognitive labels as a multi-class classification problem. The overview of the fMRI decoding model was illustrated in Figure 1.

The implementation of the GCN model was based on Tensorflow 1.12.0. The network was trained for 100 epochs with the batch size set to 128. We used Adam as the optimizer with the initial learning rate as 0.001. Additional l2 regularization of 0.0005 on weights was used to control model overfitting and the noise effect of fMRI signals. The entire dataset was split into training (70%), validation (10%), test (20%) sets using a subject-specific split scheme, which ensures that all fMRI data from the same subject was assigned to one set. The best model with the highest prediction score on the validation set was saved and then evaluated separately on the test set.

## 4 Results

### 4.1 Cognitive states can be decoded with high accuracy from 10s of fMRI data

The GCN state annotation model was evaluated using the cognitive battery of HCP task-fMRI dataset. Here we mainly focused on the 21 task conditions spanning over 6 cognitive domains, namely: emotion, language, motor, relational, social, and working memory. Using a 10 second window of fMRI time series, the 21 conditions can be identified with an average test accuracy of 89.83%, significantly different from the chance level of 4.8%. After summarizing the confusion

matrix according to the 6 cognitive domains, each cognitive domain could be identified with a recall accuracy >91%. Among the cognitive domains, the language tasks (story vs math) and motor tasks (left/right hand, left/right foot and tongue) were the most recognizable conditions, which showed the highest precision and recall scores (average f1-score = 95% and 94% respectively for language and motor conditions). Meanwhile, the relational processing and working memory task conditions were identified with the lowest performance (average f1-score = 81.6% and 84.2% respectively), with some misclassifications between emotion/relational processing and working memory tasks.

## 4.2 Classification errors due to high similarity in task stimuli

Misclassifications of cognitive states were found both between and within cognitive domains. For instance, the emotion and relational processing conditions were misidentified as working memory task, probably due to the high similarity in task stimuli. Specifically, the emotion task involves the matching of faces, overlapping with face encoding and retrieval in working memory tasks. The relational processing requires matching of shapes and textures of drawn objects, somewhat similar to encoding and retrieval of bodies and tools in working memory task. Also, some trials were misclassified within the same cognitive domain. For instance, most misclassifications within working memory task were found between 0-back and 2-back conditions, which were still observed even when the decoding model was trained for the single cognitive domain. By contrast, for face and place working memory stimuli, brain decoding reached high accuracy, regardless of using a multi-domain or single-domain classifier (misclassification less than 0.2%).

## 4.3 Performance of GCN annotation was associated with in-scanner behaviors

We found a strong association between prediction accuracy of GCN annotation and the median reaction time within scanner (Figure 2). For instance, during relational processing task which consists of two conditions, i.e. relational processing and feature matching, participants reacted faster to the matching condition than relational processing. Similarly, GCN also achieved higher prediction for matching (f1=0.96) than relational processing (f1=0.91). Moreover, within each task condition, GCN achieved higher accuracy on trials when participants were more engaged and responded faster. The analysis was performed on 200 subjects from the test set.

## 4.4 GCN achieved much better performance than SVC

We also wanted to establish if the performance of deep GCN represented a substantial improvement over more traditional machine learning tools. We thus also evaluated the same brain decoding tasks using a multi-class support vector machines classification (SVC) with a linear kernel, as our baseline model. The results showed that, using 10s of fMRI data, SVC-linear achieved much lower prediction accuracy in classifying the 21 states (89% vs 63% respectively for GCN and SVC-linear) and took a longer time for training (560s vs 9518s). Even when only focusing on a single cognitive domain, SVC-linear still showed much lower performance (96% vs 87% for motor task; 86% vs 70% for working memory conditions).

## 4.5 Saliency map demonstrate biologically meaningful features learned by GCN

We investigated whether a set of biologically meaningful features were learned by the GCN. For this purpose, we generated saliency maps on the model trained on the corresponding cognitive domain, by propagating the non-negative gradients backwards till the input layer (10). An input feature is salient or important only if its little variation causes big changes in the decoding output. Thus, high values in the saliency map indicate large contributions during prediction of cognitive states.

The two language conditions, story and mathematics, shared a number of salient features (Figure 3 A), likely related to shared cognitive processes. Both conditions involved the processing of auditory statements, which may explain high salience in the primary auditory cortex and perisylvian language-related brain regions, consisting of inferior frontal gyrus (IFG), supramarginal gyrus/angular gyrus, and superior temporal gyrus (STG). There were also some salient features found only for either mathematics or story. For instance, the story condition involved salient features in more anterior part of left IFG, including pars triangularis and orbitalis. By contrast, mathematical statements involved more posterior part, including pars opercularis of IFG and precentral sulcus. Additional inferior

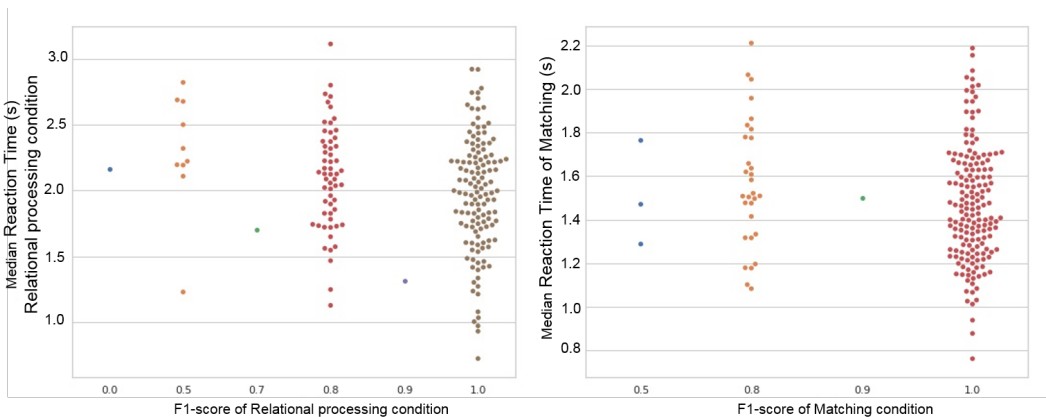

Figure 2: Association between GCN performance and in-scanner reaction time for relational processing task

temporal regions were salient for mathematics only, which have been shown to be more involved in mathematical than non-mathematical judgment tasks (1).

As expected, no salient features in the perisylvian language-related brain regions were found for the motor task (Figure 3 B). Different types of movements were associated with high salience in the primary motor and somatosensory cortex, which has been shown as the main territories engaged during movements of the human body (9). No clear somatotopic organization among different types of movements were identified here, which was somewhat expected because the primary motor and somatosensory cortex were parcellated into single strips in the Glasser's atlas (4). Some category-specific salient features were still identified, for instance in medial primary motor cortex for foot movement and in lateral orbitofrontal cortex for tongue movement.

Moreover, salient features in the ventral visual stream were identified for image recognition in working memory task (Figure 3 C). Specifically, the place stimuli activated more medial areas in the temporal cortex including parahippocampal gyrus; while the face stimuli activated more lateral temporal regions including fusiform gyrus. This observation is consistent with the known, strong spatial segregation of the neural representation for face vs place image, in fusiform face area (FFA) and parahippocampal place area (PPA) respectively (6).

In summary, the regions highlighted by the saliency maps are consistent with prior knowledge from the literature, and suggest that the GCN model has learned biologically meaningful features, rather than confounding effects for example motion artifacts.

## 5   Conclusion

We propose a new graph convolution based model to annotate human cognitive states using a short series of fMRI signals. This model annotates human brain activity with fine temporal resolution and fine cognitive granularity. Using a 10s window of fMRI signals, our model identified 21 different task conditions with a test accuracy of 89%. Besides, the GCN performance was associated with participants' in-scanner behaviors. The saliency maps of GCN also demonstrated that biologically meaningful and domain-specific features have been learned during model training. Together, our project provides an automated tool to annotate the dynamics of human cognition in real time.

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
