# OpenReview forum: "Functional Annotation of Human Cognitive States using Graph Convolution Networks"
_NeurIPS.cc/2019/Workshop/Neuro_AI — Real Neurons & Hidden Units @ NeurIPS 2019 Oral_

### Official Review · AnonReviewer2 · 2019-09-19
**Well-written and solid paper providing biologically meaningful results about neural grounding of cognitive states**

**Clarity:** 5

**Comment:**

Thorough theoretically rigorous application of a new approach for rapid decoding of cognitive states.

**Category:**

AI->Neuro

**Clarity Comment:**

Clear and well-written

**Evaluation:**

5: Excellent

**Importance:**

4: Very important

**Importance Comment:**

This paper convincingly shows how graph convolutional networks can provide very robust and interpretable predictions of cognitive states from brain activity.

**Intersection:**

4: High

**Intersection Comment:**

Nice application of GCNs for cognitive state decoding.

**Rigor Comment:**

This is simply a solid paper which includes various control analyses.

**Technical Rigor:**

4: Very convincing

---

### Official Review · AnonReviewer1 · 2019-09-27
**Interesting paper on decoding cognitive states from fMRI data with graph convolutional NNs**

**Clarity:** 4

**Comment:**

The contribution is relevant to the workshop and is well written.

**Category:**

AI->Neuro

**Clarity Comment:**

The paper is well written but is well over the limit for this workshop. Its clear that the authors didn't try to shorten their paper for this workshop.

Figure 2 says "Sates" instead of "States"

**Evaluation:**

3: Good

**Importance:**

3: Important

**Importance Comment:**

Neural decoding is an important problem with applications throughout neuroscience.

**Intersection:**

3: Medium

**Intersection Comment:**

Interesting use of ML methods to analyze neural datasets.

**Rigor Comment:**

The methods appear to be correct and are well explained.

**Technical Rigor:**

3: Convincing

---

### Decision · Program_Chairs · 2019-10-02

Accept (Oral)